# GENERATIVE ADVERSARIAL MODELS FOR LEARNING PRIVATE AND FAIR REPRESENTATIONS

## ABSTRACT

We present Generative Adversarial Privacy and Fairness (GAPF), a data-driven framework for learning private and fair representations of the data. GAPF leverages recent advances in adversarial learning to allow a data holder to learn "universal" representations that decouple a set of sensitive attributes from the rest of the dataset. Under GAPF, finding the optimal decorrelation scheme is formulated as a constrained minimax game between a generative decorrelator and an adversary. We show that for appropriately chosen adversarial loss functions, GAPF provides privacy guarantees against strong information-theoretic adversaries and enforces demographic parity. We also evaluate the performance of GAPF on multi-dimensional Gaussian mixture models and real datasets, and show how a designer can certify that representations learned under an adversary with a fixed architecture perform well against more complex adversaries.

**Keywords-** Data Privacy, Fairness, Adversarial Learning, Generative Adversarial Networks, Minimax Games, Information Theory

## 1 INTRODUCTION

The use of deep learning algorithms for data analytics has recently seen unprecedented success for a variety of problems such as image classification, natural language processing, and prediction of consumer behavior, electricity use, political preferences, to name a few. The success of these algorithms hinges on the availability of large datasets, that often contain sensitive information, and thus, may facilitate learning models that inherit societal biases leading to unintended algorithmic discrimination on legally protected groups such as race or gender. This, in turn, has led to privacy and fairness concerns and a growing body of research focused on developing representations of the dataset with fairness and/or privacy guarantees. These techniques predominantly involve designing randomizing schemes, and in recent years, distinct approaches with provable *statistical privacy or fairness* guarantees have emerged.

In the context of privacy, preserving the utility of published datasets while simultaneously providing provable privacy guarantees is a well-known challenge. While context-free privacy solutions, such as differential privacy (Dwork et al., 2006b;a; Dwork, 2008; Dwork & Roth, 2014), provide strong worst-case privacy guarantees, they often lead to a significant reduction in utility. In contrast, context-aware privacy solutions, e.g., mutual information privacy (Rebollo-Monedero et al., 2010; Calmon & Fawaz, 2012; Sankar et al., 2013; Salamatian et al., 2015; Basciftci et al., 2016), achieve improved privacy-utility tradeoff, but assume that the data holder has access to dataset statistics.

In the context of fairness, machine learning models seek to maximize predictive accuracy. Fairness concerns arise when models learned from datasets that include patterns of societal bias and discrimination inherit such biases. Thus, there is a need for actively decorrelating sensitive and non-sensitive data. In the context of publishing datasets or meaningful representations that can be "universally" used for a variety of learning tasks, modifying the training data is the most appropriate and is the focus of this work. Fairness can then be achieved by carefully designing objective functions which approximate a specific fairness definition while simultaneously ensuring maximal utility (Zemel et al., 2013; Calmon et al., 2017; Ghassami et al., 2018). This, in turn, requires dataset statistics.

Adversarial learning approaches for context-aware privacy and fairness have been studied extensively (Edwards & Storkey, 2015; Abadi & Andersen, 2016; Raval et al., 2017; Huang et al., 2017;

Tripathy et al., 2017; Beutel et al., 2017; Madras et al., 2018; Zhang et al., 2018). They allow the data curator to cleverly decorrelate the sensitive attributes from the rest of the dataset. These approaches overcome the lack of statistical knowledge by taking a *data-driven approach* that leverages recent advancements in generative adversarial networks (GANs) (Goodfellow et al., 2014; Mirza & Osindero, 2014). However, most existing efforts focus on extensive empirical studies without theoretical verification and focus predominantly on providing guarantees for a specific classification task.

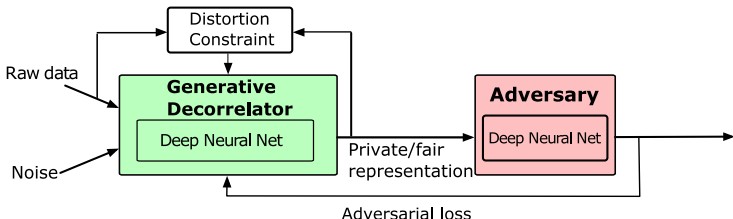

Figure 1: Generative adversarial model for privacy and fairness

This work introduces a general framework for context-aware privacy and fairness that we call *generative adversarial privacy and fairness* (GAPF) (see Figure 1). We provide precise connections to information-theoretic privacy and fairness formulations and derive game-theoretically optimal decorrelation schemes to compare against those learned directly from the data. While our framework can be generalized to learn an arbitrary representation using an encoder-decoder structure, this paper primarily focuses on learning private/fair representations of the data (of the same dimension).

**Our Contributions.** We list our main contributions below.

1. We introduce GAPF, a framework for creating private/fair representations of data using an adversarially trained conditional generative model. Unlike existing works, GAPF can create representations that are useful for a variety of classification tasks, without requiring the designer to model these tasks at training time. We validate this observation via experiments on the GENKI (Whitehill & Movellan, 2012) and HAR (Anguita et al., 2013) datasets.

2. We show that via the choice of the adversarial loss function, our framework can capture a rich class of statistical and information-theoretic adversaries. This allows us to compare data-driven approaches directly against strong inferential adversaries (e.g., a maximum *a posteriori* probability (MAP) adversary with access to dataset statistics). We also show that by carefully designing the loss functions in the GAPF framework, we can enforce demographic parity.

3. We make precise comparison between data-driven privacy/fairness methods and the minimax game-theoretic GAPF formulation. For Gaussian mixture data, we derive game-theoretically optimal decorrelation schemes and compare them with those that are directly learned in a data-driven fashion to show that the gap between theory and practice is negligible. Furthermore, we propose using mutual information estimators to verify that no adversary (regardless of their computational power) can reliably infer the sensitive attribute from the learned representation.

**Related work.** In the context of publishing datasets with privacy and utility guarantees, a number of similar approaches have been recently considered. We briefly review them here. A detailed literature review is included in Appendix A. DP-based obfuscators for data publishing have been considered in (Hamm, 2016; Liu et al., 2017). These novel approaches leverage non-generative minimax filters and deep auto-encoders to allow non-malicious entities to learn some public features from the filtered data, while preventing malicious entities from learning other sensitive features. However, DP can still incur a significant utility loss since it assumes worst-case dataset statistics. Our approach models a rich class of randomization-based schemes via a generative model that allows the generative decorrelator to tailor the noise to the dataset.

Our work is closely related to adversarial neural cryptography (Abadi & Andersen, 2016), learning censored representations (Edwards & Storkey, 2015), privacy preserving image sharing (Raval et al., 2017), privacy-preserving adversarial networks (Tripathy et al., 2017), and adversarially learning fair representation (Madras et al., 2018) in which adversarial learning is used to learn how to protect communications by encryption or hide/remove sensitive information or generate fair representation of the data. Similar to these problems, our model includes a minimax formulation and uses

adversarial neural networks to learn decorrelation schemes that prevent an adversary from inferring the sensitive variable. However, most of these papers use non-generative auto-encoders to remove sensitive information. Instead, we use a GANs-like approach to learn decorrelation schemes. We also go beyond in formulating a game-theoretic setting subject to a distortion constraint which allows us to learn private/fair representation for a variety of learning tasks. Enforcing the distortion constraint calls for a new training process that relies on the Penalty method or Augmented Lagrangian method presented in Appendix C. We show that our framework captures a rich class of statistical and information-theoretic adversaries by changing the loss function. We also compare the performance of data-driven privacy/fairness methods and the minimax game-theoretic GAPF.

Fair representations using information-theoretic objective functions and constrained optimization have been proposed in (Calmon et al., 2017; Ghassami et al., 2018). However, both approaches require the knowledge of dataset statistics, which is very difficult to obtain for real datasets. We overcome the issue of statistical knowledge by taking a *data-driven approach*, i.e., learning the representation from the data directly via adversarial models. In contrast to in-processing approaches that modify learning algorithms to ensure fair predictions (e..g, using linear programs in (Dwork et al., 2012; Fish et al., 2016) or via adversarial learning approach in (Zhang et al., 2018)), we focus on a pre-processing approach to ensure fairness for a variety of learning tasks. Using GANs to generate synthetic non-sensitive attributes and labels which ensure fairness while preserving the utility of the data (predicting the label) has been studied in (Xu et al., 2018; Sattigeri et al., 2018). Rather than using a conditional-generative model to generate synthetic data, we focus on creating fair/private representations of the original data while preserving the utility of the representations for a variety of learning tasks by learning nonlinear compression and noise adding schemes via a generative adversarial model.

## 2 GENERATIVE ADVERSARIAL MODEL FOR PRIVACY AND FAIRNESS

We consider a dataset $\mathcal{D}$ with $n$ entries where each entry is denoted as $(S, X, Y)$ where $S \in \mathcal{S}$ is the sensitive variable, $X \in \mathcal{X}$ is the public variable, and $Y \in \mathcal{Y}$ is the target (non-sensitive) variable (for learning). Instances of $X$, $S$, and $Y$ are denoted by $x$, $s$ and $y$, respectively. We assume that each entry $(X, S, Y)$ is independent and identically distributed according to $P(X, S, Y)$. Notice that we model $(X, S, Y)$ jointly in the dataset. However, GAPF does not require the knowledge of $Y$.

**Privacy and fairness.** Context-aware notions of privacy model how well an adversary, with access to the public data $X$, can infer the sensitive features $S$ from the data. Research on context-aware privacy focus on privacy that capture a range of adversarial capabilities ranging from a belief refining adversary using mutual information to quantify privacy to a guessing adversary using a hard-decision rule. On the other hand, recent results on fairness in learning applications guarantees that for a specific target variable $Y$, the prediction of a machine learning model is accurate with respect to (*w.r.t.*) $Y$ but unbiased *w.r.t.* the sensitive variable $S$. The three oft-used fairness measures are demographic parity, equalized odds, and equal opportunity. Demographic parity imposes the strongest fairness requirement via complete independence of $\hat{Y}$ and $S$, and thus, least favors (for correlated $Y$ and $S$) utility (Hardt et al., 2016). Equalized odds ensures this independence conditioned on the label $Y$ thereby ensuring equal rates for true and false positives (binary $Y$) for all demographics. Equal opportunity ensures equalized odds for the true positive case alone (Hardt et al., 2016).

When publishing a useful representation of the data for multiple users with different learning tasks, it is difficult to identify a set of target variables (labels) *a priori*. Thus, our decorrelation scheme does not restrict itself to a specific $Y$. Formally, we define the decorrelation schemes as a randomized mapping given by $\hat{X} = g(X)$. We note that $g(\cdot)$ can more generally depend on both $X$ and $S$ but for the sake of simplicity, we restrict our attention to schemes that only depend on $X$.

Let $h$ be a decision rule used by the adversary to infer the sensitive variable $S$ as $\hat{S} = h(g(X))$ from the representation $g(X)$. We allow for *hard decision rules* under which $h(g(X))$ is a direct estimate of $S$ and *soft decision rules* under which $h(g(X)) = P_h(\cdot|g(X))$ is a distribution over $\mathcal{S}$. To quantify the adversary's performance, we use a loss function $\ell(h(g(X = x)), S = s)$ defined for every public-sensitive pair $(x, s)$. Thus, the adversary's expected loss *w.r.t.* $X$ and $S$ is $L(h, g) \triangleq \mathbb{E}[\ell(h(g(X)), S)]$, where the expectation is taken over $P(X, S)$ and the randomness in $g$ and $h$.

Intuitively, the generative decorrelator would like to minimize the adversary's ability to learn $S$ reliably from the published representation. This can be trivially done by releasing an $\hat{X}$ independent of $X$. However, such an approach provides no utility for data analysts who want to learn nonsensitive variables from $\hat{X}$. To overcome this issue, we capture the loss incurred by perturbing the original data via a distortion function $d(\hat{x}, x)$, which measures how far the original data $X = x$ is from the processed data $\hat{X} = \hat{x}$. Ensuring statistical utility in turn requires constraining the average distortion $\mathbb{E}[d(g(X), X)]$ where the expectation is taken over $P(X, S)$ and the randomness in $g$.

The data holder would like to find a decorrelation scheme $g$ that is both privacy/fairness preserving (in the sense that it is difficult for the adversary to learn $S$ from $\hat{X}$) and utility preserving (in the sense that it does not distort the original data too much). In contrast, for a fixed decorrelation scheme $g$, the adversary would like to find a (potentially randomized) function $h$ that minimizes its expected loss, which is equivalent to maximizing the negative of the expected loss. This leads to a constrained minimax game between the generative decorrelator and the adversary given by

$$\min_{g(\cdot)} \max_{h(\cdot)} \quad -L(h, g), \quad s.t. \quad \mathbb{E}[d(g(X), X)] \le D, \tag{1}$$

where the constant $D \ge 0$ determines the allowable distortion for the generative decorrelator and the expectation is taken over $P(X, S)$ and the randomness in $g$ and $h$.

Our GAPF framework places no restrictions on the adversary. Indeed, different loss functions and decision rules lead to different adversarial models. In what follows, we consider a general $\alpha$-loss function $\ell(h(g(X)), s) = \frac{\alpha}{\alpha-1} \left( 1 - P_h(s|g(X))^{1-\frac{1}{\alpha}} \right), \alpha > 1$ introduced in (Liao et al., 2018). We show that $\alpha$-loss can capture various information-theoretic adversaries ranging from a hard-decision adversary under the 0-1 loss function $\ell(h(g(X)), s) = \mathbb{I}_{h(g(X)) \ne s}$ to a soft-decision adversary under the log-loss function $\ell(h(g(X)), s) = -\log P_h(s|g(X))$.

**Theorem 1.** *Under $\alpha$-loss, the optimal adversary decision rule is a '$\alpha$-tilted' conditional distribution $P_h^*(s|g(X)) = \frac{P(s|g(X))^\alpha}{\sum\limits_{s \in \mathcal{S}} P(s|g(X))^\alpha}$. The objective in equation 1 reduces to $\min\limits_{g(\cdot)} I_\alpha^a(g(X); S) - H_\alpha(S)$, where $I_\alpha^a$ is the Arimoto mutual information and $H_\alpha$ is the Rényi entropy.*

Under the hard-decision rules in which the adversary uses a 0-1 loss function, the optimal adversarial strategy simplifies to using a MAP decision rule that maximizes $P(s|g(X))$. For a soft-decision adversary under log-loss, the optimal adversarial strategy $h^*$ is $P(s|g(X))$ and the GAPF minimax problem in equation 1 simplifies to $\min_{g(\cdot)} I(g(X); S)$ subject to $\mathbb{E}[d(g(X), X)] \le D$, where $I(g(X); S)$ is the mutual information (MI) between $g(X)$ and $S$.

**Corollary 1.** *Using $\alpha$-loss, we can obtain a continuous interpolation between a hard-decision adversary under 0-1 loss ($\alpha \to \infty$) and a soft-decision adversary under log-loss function ($\alpha \to 1$).*

**Proposition 1.** *Under log-loss, GAPF enforces fairness subject to the distortion constraint. As the distortion increases, the ensuing fairness guarantee approaches ideal demographic parity.*

The proofs of Theorem 1, Corollary 1 and Proposition 1 are presented in Appendix B. Many notions of fairness rely on computing probabilities to ensure independence of sensitive and target variables that are not easy to optimize in a data-driven fashion. In Proposition 1, we propose log-loss (modeled in practice via cross-entropy) in GAPF as a proxy for enforcing fairness.

**Data-driven GAPF.** Thus far, we have focused on a setting where the data holder has access to $P(X, S)$. When $P(X, S)$ is known, the data holder can simply solve the constrained minimax optimization problem in equation 1 (game-theoretic version of GAPF) to obtain a decorrelation scheme that would perform best against a chosen type of adversary. In the absence of $P(X, S)$, we propose a data-driven version of GAPF that allows the data holder to learn decorrelation schemes directly from a dataset $\mathcal{D} = \{(x_{(i)}, s_{(i)})\}_{i=1}^n$. Under the data-driven version of GAPF, we represent the decorrelation scheme via a generative model $g(X; \theta_p)$ parameterized by $\theta_p$. This generative model takes $X$ as input and outputs $\hat{X}$. In the training phase, the data holder learns the optimal parameters $\theta_p$ by competing against a *computational adversary*: a classifier modeled by a neural network $h(g(X; \theta_p); \theta_a)$ parameterized by $\theta_a$. In the evaluation phase, the performance of the learned decorrelation scheme can be tested under a strong adversary that is computationally unbounded and has access to dataset statistics. We follow this procedure in the next section.

In theory, the functions $h$ and $g$ can be arbitrary. However, in practice, we need to restrict them to a rich hypothesis class. Figure 1 shows an example of the GAPF model in which the generative decorrelator and adversary are modeled as deep neural networks. For a fixed $h$ and $g$, if $S$ is binary, we can quantify the adversary's *empirical loss* using cross entropy $L_n(\theta_p, \theta_a) = -\frac{1}{n}\sum_{i=1}^{n} s_{(i)} \log h(g(x_{(i)}; \theta_p); \theta_a) + (1 - s_{(i)}) \log(1 - h(g(x_{(i)}; \theta_p); \theta_a))$. It is easy to generalize cross entropy to the multi-class case using the softmax function. The optimal model parameters are the solutions to

$$\min_{\theta_p} \max_{\theta_a} \quad -L_n(\theta_p, \theta_a), \quad s.t. \quad \mathbb{E}_{\mathcal{D}}[d(g(X; \theta_p), X)] \le D, \tag{2}$$

where the expectation is over $\mathcal{D}$ and the randomness in $g$.

The minimax optimization in equation 2 is a two-player non-cooperative game between the generative decorrelator and the adversary with strategies $\theta_p$ and $\theta_a$, respectively. In practice, we can learn the equilibrium of the game using an iterative algorithm (see Algorithm 1 in Appendix C). We first maximize the negative of the adversary's loss function in the inner loop to compute the parameters of $h$ for a fixed $g$. Then, we minimize the decorrelator's loss function, which is modeled as the negative of the adversary's loss function, to compute the parameters of $g$ for a fixed $h$. Observe that the distortion constraint in equation 2 makes our minimax problem different from what is extensively studied in previous works. To incorporate the distortion constraint, we use the *penalty method* (Lillo et al., 1993) to replace the constrained optimization problem by adding a penalty to the objective function. The penalty consists of a penalty parameter $\rho_t$ multiplied by a measure of violation of the constraint at the $t^{\text{th}}$ iteration. The constrained optimization problem of the generative decorrelator can be approximated by a series of unconstrained optimization problems with the loss function $L_n(\theta_p, \theta_a) + \rho_t(\max\{0, \mathbb{E}[d(g(x_{(i)}; \theta_p), x_{(i)})] - D\})^2$, where $\rho_t$ is a penalty coefficient increases with the number of iterations $t$. The algorithm and the penalty method are detailed in Appendix C.

## 3 GAPF FOR GAUSSIAN MIXTURE MODELS

To demonstrate the performance of the decorrelation schemes learned in a data-driven fashion against a computationally bounded adversary, we evaluate the performance of the learned schemes against a maximum a posteriori probability (MAP) adversary that has access to distributional information and knows the applied decorrelation schemes. First, we derive game-theoretically optimal decorrelation schemes by considering a MAP adversary. Second, we compare the game-theoretically optimal scheme against the data-driven decorrelation scheme learned from a synthetic dataset by competing against a computational adversary (modeled by a multi-layer neural network). To quantify the performance of the learned decorrelation scheme, we compute the accuracy of inferring $S$ under a MAP adversary that has access to both the joint distribution of $(X, S)$ and the decorrelation scheme. Furthermore, we use mutual information estimator (detailed in Appendix F) to demonstrate that GAPF effectively decorrelates the sensitive variables from the data. The details of the game-theoretically optimal and data-driven GAPF are included in Appendix D.

**Game-Theoretical Approach.** We focus on a setting where $S \in \{0, 1\}$ and $X$ is an $m$-dimensional Gaussian mixture random vector whose mean is dependent on $S$. Let $P(S = 1) = q$, $X|S = 0 \sim \mathcal{N}(-\mu, \Sigma)$ and $X|S = 1 \sim \mathcal{N}(\mu, \Sigma)$, where $\mu = (\mu_1, ..., \mu_m)$. We assume that $X|S = 0$ and $X|S = 1$ have the same covariance $\Sigma$. Both the generative decorrelator and the adversary have access to $P(X, S)$. In order to have a tractable model for the decorrelator, we mainly focus on linear (precisely affine) GAPF schemes $\hat{X} = g(X) = X + Z + \beta$, where $Z$ is a zero-mean multi-dimensional Gaussian random vector. This linear GAPF enables controlling both the mean and covariance of the privatized data. Although other distributions can be considered, we choose additive Gaussian noise for tractability reasons. To quantify utility of the learned representation, we use the $\ell_2$ distance between $X$ and $\hat{X}$ to obtain a distortion constraint $\mathbb{E}_{X, \hat{X}} \|X - \hat{X}\|^2 \le D$.

Without loss of generality, we assume that $\beta = (\beta_1, ..., \beta_m)$ is a constant parameter vector and $Z \sim \mathcal{N}(0, \Sigma_p)$. Let $\alpha = \sqrt{(2\mu)^T (\Sigma + \Sigma_p)^{-1} 2\mu}$, following similar analysis in (Gallager, 2013), we can show that the adversary's probability of detection is given by

$$P_d^{(G)} = qQ\left(-\frac{\alpha}{2} + \frac{1}{\alpha} \ln\left(\frac{1-q}{q}\right)\right) + (1-q)Q\left(-\frac{\alpha}{2} - \frac{1}{\alpha} \ln\left(\frac{1-q}{q}\right)\right). \tag{3}$$

**Theorem 2.** *Consider GAPF schemes $g(X) = X + Z + \beta$, where $X$ and $Z$ are multi-dimensional Gaussian random vectors with diagonal covariance matrices $\Sigma = diag(\sigma_1^2, ..., \sigma_m^2)$ and $\Sigma_p = diag(\sigma_{p_1}^2, ..., \sigma_{p_m}^2)$. The parameters of the minimax optimal decorrelation scheme are*

$$\beta_i^* = 0, \sigma_{p_i}^{*\,2} = \left( \frac{|\mu_i|}{\sqrt{\lambda_0^*}} - \sigma_i^2, 0 \right)^+, \text{ where } \lambda_0^* \text{ is chosen such that } \sum_{i=1}^m \left( \frac{|\mu_i|}{\sqrt{\lambda_0^*}} - \sigma_i^2 \right)^+ = D, \forall i =$$

$\{1, ..., m\}$. *The accuracy of the MAP adversary is given by substituting $\beta_i^*$ and $\sigma_{p_i}^*$ into equation 3.*

**Numerical Results.** Figure 2 illustrates the performance of the learned GAPF scheme against a strong theoretical MAP adversary for a 32-dimensional Gaussian mixture model with $P(S = 1) = 0.75$ and $0.5$. We observe that the inference accuracy of the MAP adversary decreases as the distortion increases and asymptotically approaches (as expected) the prior on the sensitive variable. The decorrelation scheme obtained via the data-driven approach performs very well when pitted against the MAP adversary (maximum accuracy difference around $0.7\%$ compared to the theoretical optimal). Furthermore, the estimated mutual information decreases as the distortion increases. In other words, for the data generated by Gaussian mixture model with binary sensitive variable, the data-driven version of GAPF can learn decorrelation schemes that perform as well as the decorrelation schemes computed under the theoretical version of GAPF, given that the generative decorrelator has access to the statistics of the dataset.

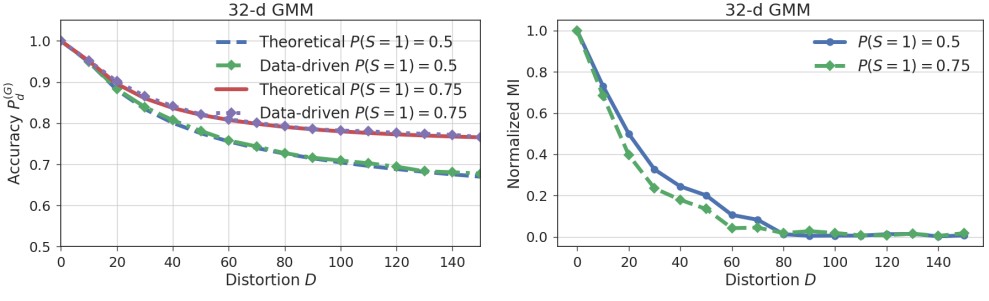

(a) Sensitive variable classification accuracy     (b) Estimated mutual information between $S$ and $\hat{X}$

Figure 2: Performance of GAPF for Gaussian mixture models

# 4 GAPF FOR REAL DATASETS

We apply our GAPF framework to real-world datasets to demonstrate its effectiveness. The GENKI dataset consists of $1,740$ training and $200$ test samples. Each data sample is a $16 \times 16$ greyscale face image with varying facial expressions. Both training and test datasets contain $50\%$ male and $50\%$ female. Among each gender, we have $50\%$ smile and $50\%$ non-smile faces. We consider gender as sensitive variable $S$ and the image pixels as public variable $X$. The HAR dataset consists of $561$ features of motion sensor data collected by a smartphone from 30 subjects performing six activities (walking, walking upstairs, walking downstairs, sitting, standing, laying). We choose subject identity as sensitive variable $S$ and features of motion sensor data as public variable $X$. The dataset is randomly partitioned into $8,000$ training and $2,299$ test samples. We train our model based on the data-driven GAPF presented in Section 2 using TensorFlow (Abadi et al., 2016).

## 4.1 GENERATIVE DECORRELATOR AND ADVERSARY MODEL

For the GENKI dataset, we consider two different decorrelator architectures: the feedforward neural network decorrelator (FNND) and the transposed convolution neural network decorrelator (TCNND). The FNND architecture uses a feedforward multi-layer neural network to combine the low-dimensional random noise $(100 \times 1)$ and the original image together (Figure 7). The TCNND takes a low-dimensional random noise and generates high-dimensional noise using a multi-layer transposed convolution neural network. The generated high-dimensional noise is added to each pixel of the original image to produce the processed image (Figure 8). For the HAR dataset, we use the FNND architecture modeled by a four-layer feedforward neural network. The details of the architectures for both the generative decorrelator and the adversary are presented in Appendix E.

| Expression | Original Data | | D = 1 | | D = 3 | | D = 5 | |
|---|---|---|---|---|---|---|---|---|
| Classification | Male | Female | Male | Female | Male | Female | Male | Female |
| False Positive Rate | 0.04 | 0.14 | 0.1 | 0.18 | 0.18 | 0.16 | 0.16 | 0.14 |
| False Negative Rate | 0.16 | 0.02 | 0.2 | 0.08 | 0.26 | 0.12 | 0.24 | 0.24 |

Table 1: Error rates for expression classification using representation learned by FNND

| Expression | Original Data | | D = 1 | | D = 3 | | D = 5 | |
|---|---|---|---|---|---|---|---|---|
| Classification | Male | Female | Male | Female | Male | Female | Male | Female |
| False Positive Rate | 0.04 | 0.14 | 0.04 | 0.16 | 0.06 | 0.12 | 0.08 | 0.16 |
| False Negative Rate | 0.16 | 0.02 | 0.2 | 0.08 | 0.2 | 0.14 | 0.18 | 0.16 |

Table 2: Error rates for expression classification using representation learned by TCNND

## 4.2 ILLUSTRATION OF RESULTS

**The GENKI Dataset.** Figure 3a illustrates the gender classification accuracy of the adversary for different values of distortion. It can be seen that the adversary's accuracy of classifying the sensitive variable (gender) decreases progressively as the distortion increases. Given the same distortion value, FNND achieves lower gender classification accuracy compared to TCNND. An intuitive explanation is that the FNND uses both the noise vector and the original image to generate the processed image. However, the TCNND generates the noise mask that is independent of the original image pixels and adds the noise mask to the original image in the final step. To demonstrate the effectiveness of the learned GAPF schemes, we compare the gender classification accuracy of the learned GAPF schemes with adding uniform or Laplace noise. Figure 3a shows that for the same distortion, the learned GAPF schemes achieve much lower gender classification accuracies than using uniform or Laplace noise. Furthermore, the estimated mutual information $\hat{I}(\hat{X}; S)$ normalized by $\hat{I}(X; S)$ also decreases as the distortion increases (Figure 3b).

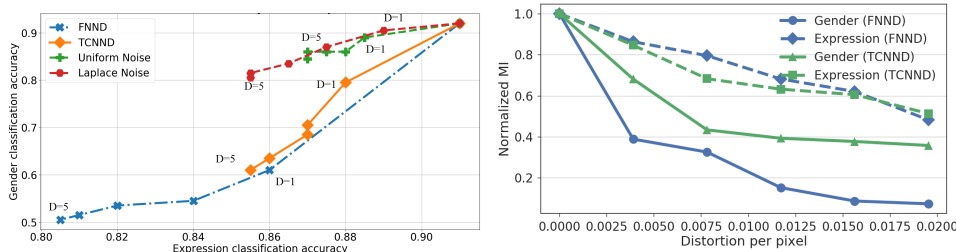

(a) Gender vs. expression classification accuracy  (b) Normalized mutual information estimation

Figure 3: Privacy/fairness-utility tradeoff and mutual information estimation for GENKI

To evaluate the influence of GAPF on other non-sensitive variable $(Y)$ classification tasks, we train another CNN (see Figure 9) to perform facial expression classification on datasets processed by different decorrelation schemes. The trained model is then tested on the original test data. In Figure 3a, we observe that the expression classification accuracy decreases gradually as the distortion increases. Even for a large distortion value (5 per image), the expression classification accuracy only decreases by $10\%$. Furthermore, the estimated normalized mutual information $\hat{I}(\hat{X}; Y)/\hat{I}(X; Y)$ decreases much slower than $\hat{I}(\hat{X}; S)/\hat{I}(X; S)$ as the distortion increases (Figure 3b).

Table 1 and 2 present different error rates for the facial expression classifiers trained using data representations created by different decorrelator architectures. We observe that as distortion increases, the error rates difference for different sensitive groups decrease. This implies the classifier's decision is less biased to the sensitive variables when trained using the processed data. When $D = 5$, the differences are already very small. Furthermore, we notice that the FNND architecture performs better in enforcing fairness but suffers from higher error rate. The images processed by FNND is shown in Figure 4. The decorrelator changes mostly eyes, nose, mouth, beard, and hair.

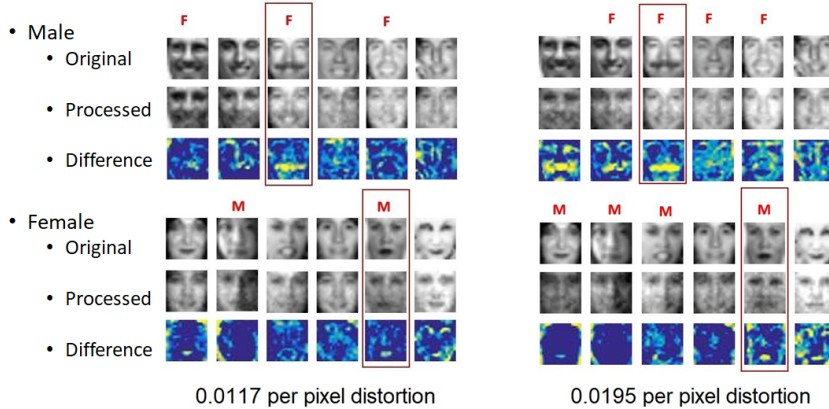

Figure 4: Perturbed images with different per pixel distortion using FNND

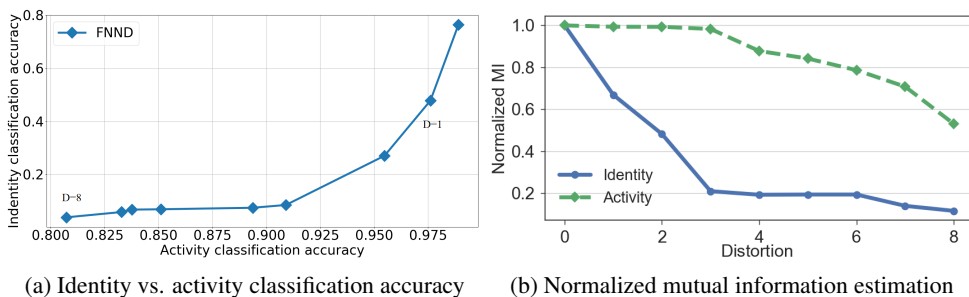

(a) Identity vs. activity classification accuracy     (b) Normalized mutual information estimation

Figure 5: Privacy/fairness-utility tradeoff and mutual information estimation for HAR

**The HAR Dataset.** Figure 5a illustrates the activity and identity classification accuracy for different values of distortion. The adversary's sensitive variable (identity) classification accuracy decreases progressively as the distortion increases. When the distortion is small ($D = 2$), the adversary's classification accuracy is already around $27\%$. If we increase the distortion to $8$, the classification accuracy further decreases to $3.8\%$. Figure 5a depicts that even for a large distortion value ($D = 8$), the activity classification accuracy only decreases by $18\%$ at most. Furthermore, Figure 5b shows that the estimated normalized mutual information also decreases as the distortion increases.

## 5 CONCLUSION

We have introduced a novel adversarial learning framework for creating private/fair representations of the data with verifiable guarantees. GAPF allows the data holder to learn the decorrelation scheme directly from the dataset (to be published) without requiring access to dataset statistics. Under GAPF, finding the optimal decorrelation scheme is formulated as a game between two players: a generative decorrelator and an adversary. We have shown that for appropriately chosen loss functions, GAPF can provide guarantees against strong information-theoretic adversaries, such as MAP and MI adversaries. It can also enforce fairness, quantified via demographic parity by using the log-loss function. We have also validated the performance of GAPF on Gaussian mixture models and real datasets. There are several fundamental questions that we seek to address. An immediate one is to develop techniques to rigorously benchmark data-driven results for large datasets against computable theoretical guarantees. More broadly, it will be interesting to investigate the robustness and convergence speed of the decorrelation schemes learned in a data-driven fashion. In this paper, we connect our objective function in GAPF with demographic parity. Since there is no single metric for fairness, this leaves room for designing objective functions that link to other fairness metrics such as equalized odds and equal opportunity.

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

# Supplementary Material

## A   LITERATURE REVIEW

In the context of publishing datasets with privacy and utility guarantees, a number of similar approaches have been recently considered. We briefly review them and clarify how our work is different. DP-based obfuscators for data publishing have been considered in (Hamm, 2016; Liu et al., 2017). The author in (Hamm, 2016) considers a deterministic, compressive mapping of the input data with differentially private noise added either before or after the mapping. The approach in (Liu et al., 2017) relies on using deep auto-encoders to determine the relevant feature space to add differentially private noise, thereby eliminating the need to add noise to the original data. These novel approaches leverage minimax filters and deep auto-encoders to allow non-malicious entities to learn some public features from the filtered data, while preventing malicious entities from learning other sensitive features. Both approaches incorporate a notion of context-aware privacy and achieve better privacy-utility tradeoffs while using DP to enforce privacy. However, DP can still incur a significant utility loss since it assumes worst-case dataset statistics. Our approach models a rich class of randomization-based schemes via a generative model that allows the generative decorrelator to tailor the noise to the dataset.

Our work is closely related to adversarial neural cryptography (Abadi & Andersen, 2016), learning censored representations (Edwards & Storkey, 2015), privacy preserving image sharing (Raval et al., 2017), privacy-preserving adversarial networks (Tripathy et al., 2017), and adversarially learning fair representation (Madras et al., 2018) in which adversarial learning is used to learn how to protect communications by encryption or hide/remove sensitive information or generate fair representation of the data. Similar to these problems, our model includes a minimax formulation and uses adversarial neural networks to learn decorrelation schemes. However, in (Edwards & Storkey, 2015; Raval et al., 2017; Madras et al., 2018), the authors use non-generative auto-encoders to remove sensitive information. Instead, we use a GANs-like approach to learn decorrelation schemes that prevent an adversary from inferring the sensitive variable. Furthermore, these formulations uses weighted combination of different loss functions to balance privacy with utility. We also go beyond in formulating a game-theoretic setting subject to a distortion constraint. These approaches are not equivalent because of the non-convexity (resp. concavity) of the minimax problem with respect to the decorrelator (resp. adversary) neural network parameters and requires new methods to enforce the distortion constraint during the training process. The distortion constraint allows us to directly limit the amount of distortion added to learn the private/fair representation for a variety of learning tasks, which is crucial for preserving the utility of the learned representation. Moreover, we compare the performance of the decorrelation schemes learned in an adversarial fashion with the game-theoretically optimal ones for canonical synthetic data models thereby providing formal verification of decorrelation schemes that are learned by competing against computational adversaries. Finally, we propose using mutual information as a criterion to certify that the representations we learned adversarially against an attacker with a fixed architecture generalize against unseen attackers with (possibly) more complex architecture.

Fair representations using information-theoretic objective functions and constrained optimization have been proposed in (Calmon et al., 2017; Ghassami et al., 2018). However, both approaches require the knowledge of dataset statistics, which are very difficult to obtain for real datasets. We overcome the issue of statistical knowledge by taking a *data-driven approach*, i.e., learning the representation from the data directly via adversarial models. In contrast to in-processing approaches that modify learning algorithms to ensure fair predictions (e..g, using linear programs in (Dwork et al., 2012; Fish et al., 2016) or via adversarial learning approach in (Zhang et al., 2018)), we focus on a pre-processing approach to ensure fairness for a variety of learning tasks.

Generative adversarial networks (GANs) have recently received a lot of attention in the machine learning community (Goodfellow et al., 2014; Mirza & Osindero, 2014). Ultimately, deep generative models hold the promise of discovering and efficiently internalizing the statistics of the target signal to be generated. Using GANs to generate synthetic non-sensitive attributes and labels which ensure fairness while preserving the utility of the data (predicting the label) has been studied in (Xu et al., 2018; Sattigeri et al., 2018). The goal here is to develop a conditional GAN-based model to ensure fairness in the system by learning to generate a fairer synthetic dataset using an unconstrained

minimax game with carefully designed loss functions corresponding with both fairness and utility. The synthetic data is generated by a conditional generative adversarial network (GAN) which generates the non-sensitive attributes-label pair given the noise variable and the sensitive attribute. The utility is preserved by generating data that is very similar to the original data. To ensure fairness, the generator generates data samples such that an auxiliary classifier trained to predict the sensitive attribute from the synthetic data performs as poorly as possible. The methods presented in these papers are very different from our method since we are focusing on creating a fair/private representations of the original data while preserving the utility of the representation for a variety of learning tasks. There are different ways for enforcing fairness, and our work presents a framework that aids in achieving this goal. More work is needed to be done in this area.

## B  THEORETICAL RESULTS OF GAPF

Our GAPF framework places no restrictions on the adversary. Indeed, different loss functions and decision rules lead to different adversarial models. In what follows, we will discuss a variety of loss functions under hard and soft decision rules, and show how our GAPF framework can recover several popular information theoretic privacy notions. We will also show that we can obtain a continuous interpolation between a hard-decision adversary under 0-1 loss function and a soft-decision adversary under log-loss function using the $\alpha$-loss function.

**Hard Decision Rules.**  When the adversary adopts a hard decision rule, $h(g(X))$ is an estimate of $S$. Under this setting, we can choose $\ell(h(g(X)), S)$ in a variety of ways. For instance, if $S$ is continuous, the adversary can attempt to minimize the difference between the estimated and true sensitive variable values. This can be achieved by considering a squared loss function

$$\ell(h(g(X)), S) = (h(g(X)) - S)^2, \tag{4}$$

which is known as the $\ell_2$ loss. In this case, one can verify that the adversary's optimal decision rule is $h^* = \mathbb{E}[S|g(X)]$, which is the conditional mean of $S$ given $g(X)$. Furthermore, under the adversary's optimal decision rule, the minimax problem in equation 1 simplifies to

$$\min_{g(\cdot)} -\mathrm{mmse}(S|g(X)) = -\max_{g(\cdot)} \mathrm{mmse}(S|g(X)),$$

subject to the distortion constraint. Here $\mathrm{mmse}(S|g(X))$ is the resulting minimum mean square error (MMSE) under $h^* = \mathbb{E}[S|g(X)]$. Thus, under the $\ell_2$ loss, GAPF provides privacy guarantees against an MMSE adversary. On the other hand, when $S$ is discrete (e.g., age, gender, political affiliation, etc), the adversary can attempt to maximize its classification accuracy. This is achieved by considering a 0-1 loss function (Nguyen & Sanner, 2013) given by

$$\ell(h(g(X)), S) = \begin{cases} 0 & \text{if } h(g(X)) = S \\ 1 & \text{otherwise} \end{cases}. \tag{5}$$

In this case, one can verify that the adversary's optimal decision rule is the *maximum a posteriori probability* (MAP) decision rule: $h^* = \arg\max_{s \in \mathcal{S}} P(s|g(X))$, with ties broken uniformly at random. Moreover, under the MAP decision rule, the minimax problem in equation 1 reduces to

$$\min_{g(\cdot)} -(1 - \max_{s \in \mathcal{S}} P(s, g(X))) = \min_{g(\cdot)} \max_{s \in \mathcal{S}} P(s, g(X)) - 1, \tag{6}$$

subject to the distortion constraint. Thus, under a 0-1 loss function, the GAPF formulation provides privacy guarantees against a MAP adversary.

**Soft Decision Rules.**  Instead of a *hard decision* rule, we can also consider a broader class of *soft decision* rules where $h(g(X))$ is a distribution over $\mathcal{S}$; i.e., $h(g(X)) = P_h(s|g(X))$ for $s \in \mathcal{S}$. In this context, we can analyze the performance under a log-loss

$$\ell(h(g(X)), s) = \log \frac{1}{P_h(s|g(X))}. \tag{7}$$

In this case, the objective of the adversary simplifies to

$$\max_{h(\cdot)} -\mathbb{E}[\log \frac{1}{P_h(s|g(X))}] = -H(S|g(X)),$$

and that the maximization is attained at $P_h^*(s|g(X)) = P(s|g(X))$. Therefore, the optimal adversarial decision rule is determined by the true conditional distribution $P(s|g(X))$, which we assume is known to the data holder in the game-theoretic setting. Thus, under the log-loss function, the minimax optimization problem in equation 1 reduces to

$$\min_{g(\cdot)} -H(S|g(X)) = \min_{g(\cdot)} I(g(X); S) - H(S),$$

subject to the distortion constraint. Thus, under the log-loss in equation 7, GAPF is equivalent to using MI as the privacy metric (Calmon & Fawaz, 2012).

The 0-1 loss captures a strong guessing adversary; in contrast, log-loss or information-loss models a belief refining adversary.

### B.1 PROOF OF THEOREM 1

Consider the $\alpha$-loss function (Liao et al., 2018)

$$\ell(h(g(X)), s) = \frac{\alpha}{\alpha - 1} \left( 1 - P_h(s|g(X))^{1 - \frac{1}{\alpha}} \right), \tag{8}$$

for any $\alpha > 1$. Denoting $H_\alpha^{\mathrm{a}}(S|g(X))$ as the Arimoto conditional entropy of order $\alpha$, one can verify that

$$\max_{h(\cdot)} -\mathbb{E}\left[ \frac{\alpha}{\alpha - 1} \left( 1 - P_h(s|g(X))^{1 - \frac{1}{\alpha}} \right) \right] = -H_\alpha^{\mathrm{a}}(S|g(X)),$$

which is achieved by a '$\alpha$-tilted' conditional distribution

$$P_h^*(s|g(X)) = \frac{P(s|g(X))^\alpha}{\sum\limits_{s \in \mathcal{S}} P(s|g(X))^\alpha}.$$

Under this choice of a decision rule, the objective of the minimax optimization in equation 1 reduces to

$$\min_{g(\cdot)} -H_\alpha^{\mathrm{a}}(S|g(X)) = \min_{g(\cdot)} I_\alpha^{\mathrm{a}}(g(X); S) - H_\alpha(S), \tag{9}$$

where $I_\alpha^{\mathrm{a}}$ is the Arimoto mutual information and $H_\alpha$ is the Rényi entropy.

### B.2 PROOF OF COROLLARY 1

For large $\alpha$ ($\alpha \to \infty$), this loss approaches that of the 0-1 (MAP) adversary in the limit. As $\alpha$ decreases, the convexity of the loss function encourages the estimator $\hat{S}$ to be probabilistic, as it increasingly rewards correct inferences of lesser and lesser likely outcomes (in contrast to a hard decision rule by a MAP adversary of the most likely outcome) conditioned on the revealed data. As $\alpha \to 1$, equation 8 yields the logarithmic loss, and the optimal belief $P_{\hat{S}}$ is simply the posterior belief. Therefore, using $\alpha$-loss, we can obtain a continuous interpolation between a hard-decision adversary under 0-1 loss ($\alpha \to \infty$) and a soft-decision adversary under log-loss function ($\alpha \to 1$).

### B.3 PROOF OF PROPOSITION 1

Let's consider an arbitrary target variable $Y$ which a user is interested in learning from the data. The objective of the learning task is to train a good model that takes $\hat{X}$ to predict $Y$. Thus, we have the Markov chain: $S \to X \to \hat{X} \to \hat{Y}$, where $\hat{Y}$ is an estimate of $Y$ from the trained machine learning model. According to data processing inequality, we have $I(S; \hat{X}) \geq I(S; \hat{Y})$. As we have shown in the above analysis, for the log-loss function, the objective of GAPF is equivalent to minimizing $I(S; \hat{X})$, which is an upperbound on $I(S; \hat{Y})$. Notice that demographic parity requires $S$ and $\hat{Y}$ to be independent, which is equivalent to $I(S; \hat{Y}) = 0$. Since mutual information is non-negative, GAPF ensures fairness by minimizing an upperbound of $I(S; \hat{Y})$ subject to the distortion constraint under the log-loss function. As the distortion increases, the ensuing fairness guarantee approaches ideal demographic parity by enforcing $I(S; \hat{Y}) \leq I(S; \hat{X}) = 0$.

## C    ALTERNATE MINIMAX ALGORITHM

In this section, we present the alternate minimax algorithm to learn the GAPF scheme from a dataset. The alternating minimax privacy preserving algorithm is presented in Algorithm 1. To incorporate

---

**Algorithm 1** Alternating minimax privacy preserving algorithm

---

*Input:* dataset $\mathcal{D}$, distortion parameter $D$, iteration number $T$

*Output:* Optimal generative decorrelator parameter $\theta_p$

**procedure** ALERNATE MINIMAX($\mathcal{D}, D, T$)

   Initialize $\theta_p^1$ and $\theta_a^1$

   **for** $t = 1, ..., T$ **do**

      Random minibatch of $M$ datapoints $\{x_{(1)}, ..., x_{(M)}\}$ drawn from full dataset

      Generate $\{\hat{x}_{(1)}, ..., \hat{x}_{(M)}\}$ via $\hat{x}_{(i)} = g(x_{(i)}, s_{(i)}; \theta_p^t)$

      Update the adversary parameter $\theta_a^{t+1}$ by stochastic gradient ascend for $j$ epochs

$$\theta_a^{t+1} = \theta_a^t + \alpha_t \nabla_{\theta_a^t} \frac{1}{M} \sum_{i=1}^{M} -\ell(h(\hat{x}_{(i)}; \theta_a^t), s_{(i)}), \quad \alpha_t > 0$$

      Compute the descent direction $\nabla_{\theta_p^t} l(\theta_p^t, \theta_a^{t+1})$, where

$$\ell(\theta_p^t, \theta_a^{t+1}) = -\frac{1}{M} \sum_{i=1}^{M} \ell(h(g(x_{(i)}, s_{(i)}; \theta_p^t); \theta_a^{t+1}), s_{(i)})$$

   subject to $\frac{1}{M} \sum_{i=1}^{M} [d(g(x_{(i)}, s_{(i)}; \theta_p^t), x_{(i)})] \leq D$

      Perform line search along $\nabla_{\theta_p^t} l(\theta_p^t, \theta_a^{t+1})$ and update

$$\theta_p^{t+1} = \theta_p^t - \alpha_t \nabla_{\theta_p^t} \ell(\theta_p^t, \theta_a^{t+1})$$

      Exit if solution converged

   **return** $\theta_p^{t+1}$

---

the distortion constraint into the learning algorithm, we use the *penalty method* (Lillo et al., 1993) and *augmented Lagrangian method* (Eckstein & Yao, 2012) to replace the constrained optimization problem by a series of unconstrained problems whose solutions asymptotically converge to the solution of the constrained problem. Under the penalty method, the unconstrained optimization problem is formed by adding a penalty to the objective function. The added penalty consists of a penalty parameter $\rho_t$ multiplied by a measure of violation of the constraint. The measure of violation is non-zero when the constraint is violated and is zero if the constraint is not violated. Therefore, in Algorithm 1, the constrained optimization problem of the decorrelator can be approximated by a series of unconstrained optimization problems with the loss function

$$\ell(\theta_p^t, \theta_a^{t+1}) = -\frac{1}{M} \sum_{i=1}^{M} \ell(h(g(x_{(i)}; \theta_p^t); \theta_a^{t+1}), s_{(i)}) + \rho_t (\max\{0, \frac{1}{M} \sum_{i=1}^{M} d(g(x_{(i)}; \theta_p^t), x_{(i)}) - D\})^2,$$

$$(10)$$

where $\rho_t$ is a penalty coefficient which increases with the number of iterations $t$. For convex optimization problems, the solution to the series of unconstrained problems will eventually converge to the solution of the original constrained problem (Lillo et al., 1993).

The augmented Lagrangian method is another approach to enforce equality constraints by penalizing the objective function whenever the constraints are not satisfied. Different from the penalty method, the augmented Lagrangian method combines the use of a Lagrange multiplier and a quadratic penalty term. Note that this method is designed for equality constraints. Therefore, we introduce a slack variable $\delta$ to convert the inequality distortion constraint into an equality constraint. Using the augmented Lagrangian method, the constrained optimization problem of the decorrelator can be replaced by a series of unconstrained problems with the loss function given by

$$\ell(\theta_p^t, \theta_a^{t+1}, \delta) = -\frac{1}{M}\sum_{i=1}^{M}\ell(h(g(x_{(i)};\theta_p^t);\theta_a^{t+1}), s_{(i)}) + \frac{\rho_t}{2}(\frac{1}{M}\sum_{i=1}^{M}d(g(x_{(i)};\theta_p^t), x_{(i)}) + \delta - D)^2$$

(11)

$$- \lambda_t(\frac{1}{M}\sum_{i=1}^{M}d(g(x_{(i)};\theta_p^t), x_{(i)}) + \delta - D),$$

where $\rho_t$ is a penalty coefficient which increases with the number of iterations $t$ and $\lambda_t$ is updated according to the rule $\lambda_{t+1} = \lambda_t - \rho_t(\frac{1}{M}\sum_{i=1}^{M}d(g(x_{(i)};\theta_p^t), x_{(i)}) + \delta - D)$. For convex optimization problems, the solution to the series of unconstrained problems formulated by the augmented Lagrangian method also converges to the solution of the original constrained problem (Eckstein & Yao, 2012).

## D    GAPF FOR GAUSSIAN MIXTURE MODELS

### D.1    PROOF OF THEOREM 2

*Proof.* Since $\mathbb{E}_{X,\hat{X}}[d(\hat{X}, X)] = \mathbb{E}_{X,\hat{X}}\|X - \hat{X}\|^2 = \mathbb{E}\|Z + \beta\|^2 = \|\beta\|^2 + tr(\Sigma_p)$, the distortion constraint implies that $\|\beta\|^2 + tr(\Sigma_p) \leq D$. Let us consider $\hat{X} = X + Z + \beta$, where $\beta \in \mathbb{R}$ and $\Sigma_p$ is a diagonal covariance whose diagonal entries is given by $\{\sigma_{p_1}^2, ..., \sigma_{p_m}^2\}$. Given the MAP adversary's optimal inference accuracy in equation 3, the objective of the decorrelator is to

$$\min_{\beta, \Sigma_p} \quad P_d^{(G)}$$ (12)

$$s.t. \quad \|\beta\|^2 + tr(\Sigma_p) \leq D.$$

Define $\frac{1-q}{q} = \eta$. The gradient of $P_d^{(G)}$ *w.r.t.* $\alpha$ is given by

$$\frac{\partial P_d^{(G)}}{\partial \alpha} = \tilde{p}\left(-\frac{1}{\sqrt{2\pi}}e^{-\frac{\left(-\frac{\alpha}{2}+\frac{1}{\alpha}\ln\eta\right)^2}{2}}\right)\left(-\frac{1}{2} - \frac{1}{\alpha^2}\ln\eta\right)$$ (13)

$$+ (1-\tilde{p})\left(-\frac{1}{\sqrt{2\pi}}e^{-\frac{\left(-\frac{\alpha}{2}-\frac{1}{\alpha}\ln\eta\right)^2}{2}}\right)\left(-\frac{1}{2} + \frac{1}{\alpha^2}\ln\eta\right)$$

$$= \frac{1}{2\sqrt{2\pi}}\left(\tilde{p}e^{-\frac{\left(-\frac{\alpha}{2}+\frac{1}{\alpha}\ln\eta\right)^2}{2}} + (1-\tilde{p})e^{-\frac{\left(-\frac{\alpha}{2}-\frac{1}{\alpha}\ln\eta\right)^2}{2}}\right)$$ (14)

$$+ \frac{\ln\eta}{\alpha^2\sqrt{2\pi}}\left(\tilde{p}e^{-\frac{\left(-\frac{\alpha}{2}+\frac{1}{\alpha}\ln\eta\right)^2}{2}} - (1-\tilde{p})e^{-\frac{\left(-\frac{\alpha}{2}-\frac{1}{\alpha}\ln\eta\right)^2}{2}}\right).$$

Note that

$$\frac{\tilde{p}e^{-\frac{\left(-\frac{\alpha}{2}+\frac{1}{\alpha}\ln\eta\right)^2}{2}}}{(1-\tilde{p})e^{-\frac{\left(-\frac{\alpha}{2}-\frac{1}{\alpha}\ln\eta\right)^2}{2}}} = \frac{\tilde{p}}{1-\tilde{p}}e^{\frac{\left(-\frac{\alpha}{2}-\frac{1}{\alpha}\ln\eta\right)^2 - \left(-\frac{\alpha}{2}+\frac{1}{\alpha}\ln\eta\right)^2}{2}} = \frac{\tilde{p}}{1-\tilde{p}}e^{\frac{2\ln\eta}{2}} = \frac{\tilde{p}}{1-\tilde{p}}e^{\ln\eta} = 1.$$

(15)

Therefore, the second term in equation 14 is 0. Furthermore, the first term in equation 14 is always positive. Thus, $P_{\mathrm{d}}^{(\mathrm{G})}$ is monotonically increasing in $\alpha$. As a result, the optimization problem in equation 12 is equivalent to

$$\min_{\beta, \Sigma_p} \quad (2\mu)^T (\Sigma + \Sigma_p)^{-1} 2\mu \tag{16}$$

$$s.t. \quad \|\beta\|^2 + tr(\Sigma_p) \leq D.$$

The objective function in equation 16 can be written as

$$2\begin{bmatrix} \mu_1 & \mu_2 & \cdots & \mu_m \end{bmatrix} \begin{bmatrix} \frac{1}{\sigma_1^2 + \sigma_{p_1}^2} & 0 & \cdots & 0 \\ 0 & \frac{1}{\sigma_2^2 + \sigma_{p_2}^2} & \cdots & 0 \\ \vdots & \vdots & \ddots & \vdots \\ 0 & 0 & \cdots & \frac{1}{\sigma_m^2 + \sigma_{p_m}^2} \end{bmatrix} 2 \begin{bmatrix} \mu_1 \\ \mu_2 \\ \vdots \\ \mu_m \end{bmatrix} = \sum_{i=1}^m \frac{4\mu_i^2}{\sigma_i^2 + \sigma_{p_i}^2}.$$

Thus, the optimization problem in equation 16 is equivalent to

$$\min_{\beta, \sigma_{p_1}^2, \ldots, \sigma_{p_m}^2} \quad \sum_{i=1}^m \frac{\mu_i^2}{\sigma_i^2 + \sigma_{p_i}^2} \tag{17}$$

$$s.t. \quad \|\beta\|^2 + tr(\Sigma_p) \leq D$$

$$\sigma_{p_i}^2 \geq 0 \quad \forall i \in \{1, 2, \ldots m\}.$$

Since a non-zero $\beta$ does not affect the objective function but result in positive distortion, the optimal scheme satisfies $\beta = (0, \ldots, 0)$. Furthermore, the Lagrangian of the above optimization problem is given by

$$L(\sigma_{p_1}^2, \ldots, \sigma_{p_m}^2, \lambda) = \sum_{i=1}^m \frac{\mu_i^2}{\sigma_i^2 + \sigma_{p_i}^2} + \lambda_0 \left( \sum_{i=1}^m \sigma_{p_i}^2 - D \right) - \sum_{i=1}^m \lambda_i \sigma_{p_i}^2, \tag{18}$$

where $\lambda = \{\lambda_0, \ldots, \lambda_m\}$ denotes the Lagrangian multipliers associated with the constraints. Taking the derivatives of $L(\sigma_{p_1}^2, \ldots, \sigma_{p_m}^2, \lambda)$ with respect to $\sigma_{p_i}^2, \forall i \in \{1, \ldots, m\}$, we have

$$\frac{\partial L(\sigma_{p_1}^2, \ldots, \sigma_{p_m}^2, \lambda)}{\partial \sigma_{p_i}^2} = -\frac{\mu_i^2}{(\sigma_i^2 + \sigma_{p_i}^2)^2} + \lambda_0 - \lambda_i. \tag{19}$$

Notice that the objective function in equation 16 is decreasing in $\sigma_{p_i}^2, \forall i \in \{1, \ldots, m\}$. Thus, the optimal solution $\sigma_{p_i}^{*\,2}$ satisfies $\sum_{i=1}^m \sigma_{p_i}^{*\,2} = D$. By the KKT conditions, we have

$$\left. \frac{\partial L(\sigma_{p_1}^2, \ldots, \sigma_{p_m}^2, \lambda)}{\partial \sigma_{p_i}^2} \right|_{\sigma_{p_i}^2 = \sigma_{p_i}^{*\,2}, \lambda = \lambda^*} = -\frac{\mu_i^2}{(\sigma_i^2 + \sigma_{p_i}^{*\,2})^2} + \lambda_0^* - \lambda_i^* = 0. \tag{20}$$

Since $\lambda_i^*, i \in \{0, 1, \ldots, m\}$ is dual feasible, we have $\lambda_i^* \geq 0, i \in \{0, 1, \ldots, m\}$. Therefore

$$\lambda_0^* \geq \frac{\mu_i^2}{(\sigma_i^2 + \sigma_{p_i}^{*\,2})^2}.$$

If $\lambda_0^* > \frac{\mu_i^2}{\sigma_i^4}$, we have $\lambda_0^* > \frac{\mu_i^2}{(\sigma_i^2 + \sigma_{p_i}^{*\,2})^2}$. This implies $\lambda_i^* > 0$. Thus, by complementary slackness, $\sigma_{p_i}^{*\,2} = 0$. On the other hand, if $\lambda_0^* < \frac{\mu_i^2}{\sigma_i^4}$, we have $\sigma_{p_i}^{*\,2} > 0$. Furthermore, by the complementary slackness condition, $\lambda_i^* \sigma_{p_i}^{*\,2} = 0, \forall \sigma_{p_i}^{*\,2}$. This implies $\lambda_i^* = 0, \forall \sigma_{p_i}^{*\,2} > 0$. As a result, for all $\sigma_{p_i}^{*\,2} > 0$, we have

$$\frac{|\mu_i|}{\sqrt{\lambda_0^*}} = \sigma_i^2 + \sigma_{p_i}^{*\,2}. \tag{21}$$

Therefore, $\sigma_{p_i}^{*\,2} = \max\{\frac{|\mu_i|}{\sqrt{\lambda_0^*}} - \sigma_i^2, 0\} = \left(\frac{|\mu_i|}{\sqrt{\lambda_0^*}} - \sigma_i^2\right)^+$ with $\sum_{i=1}^m \sigma_{p_i}^{*\,2} = D$. Substitute this optimal solution into equation 3 with $\alpha = \sqrt{(2\mu)^T (\Sigma + \Sigma_p)^{-1} 2\mu}$, we obtain the accuracy of the MAP adversary. $\qquad \square$

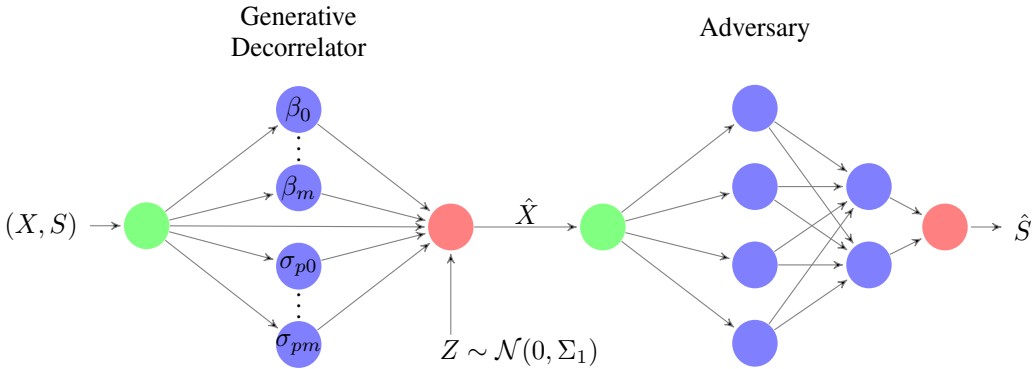

Figure 6: Neural network structure of linear GAPF for Gaussian mixture data

We observe that the when $\sigma_i^2$ is greater than some threshold $\frac{|\mu_i|}{\sqrt{\lambda_0^*}}$, no noise is added to the data on this dimension due to the high variance. When $\sigma_i^2$ is smaller than $\frac{|\mu_i|}{\sqrt{\lambda_0^*}}$, the amount of noise added to this dimension is proportional to $|\mu_i|$; this is intuitive since a large $|\mu_i|$ indicates the two conditionally Gaussian distributions are further away on this dimension, and thus, distinguishable. Thus, more noise needs to be added in order to reduce the MAP adversary's inference accuracy.

### D.2 DATA-DRIVEN APPROACH

For the data-driven linear GAPF scheme, we assume the generative decorrelator only has access to the dataset $\mathcal{D}$ with $n$ data samples but not the actual distribution of $(X, S)$. Computing the optimal decorrelation scheme becomes a learning problem. In the training phase, the data holder learns the parameters of the GAPF scheme by competing against a computational adversary modeled by a multi-layer neural network. When convergence is reached, we evaluate the performance of the learned scheme by comparing with the one obtained from the game-theoretic approach. To quantify the performance of the learned GAPF scheme, we compute the accuracy of inferring $S$ under a strong MAP adversary that has access to both the joint distribution of $(X, S)$ and the decorrelation scheme.

Since the sensitive variable $S$ is binary, we measure the training loss of the adversary network by the empirical log-loss function

$$L_n(\theta_p, \theta_a) = -\frac{1}{n} \sum_{i=1}^{n} s_{(i)} \log h(g(x_{(i)}; \theta_p); \theta_a) + (1 - s_{(i)}) \log(1 - h(g(x_{(i)}; \theta_p); \theta_a)). \quad (22)$$

For a fixed decorrelator parameter $\theta_p$, the adversary learns the optimal $\theta_a^*$ by maximizing equation 22. For a fixed $\theta_a$, the decorrelator learns the optimal $\theta_p^*$ by minimizing $-L_n(h(g(X; \theta_p); \theta_a), S)$ subject to the distortion constraint $\mathbb{E}_{X, \hat{X}} \|X - \hat{X}\|^2 \leq D$.

As shown in Figure 6, the decorrelator is modeled by a two-layer neural network with parameters $\theta_p = \{\beta_0, ..., \beta_m, \sigma_{p0}, ..., \sigma_{pm}\}$, where $\beta_k$ and $\sigma_{pk}$ represent the mean and standard deviation for each dimension $k \in \{1, ..., m\}$, respectively. The random noise $Z$ is drawn from a $m$-dimensional independent zero-mean standard Gaussian distribution with covariance $\Sigma_1$. Thus, we have $\hat{X}_k = X_k + \beta_k + \sigma_{pk} Z_k$. The adversary, whose goal is to infer $S$ from privatized data $\hat{X}$, is modeled by a three-layer neural network classifier with leaky ReLU activations.

To incorporate the distortion constraint into the learning process, we add a penalty term to the objective of the decorrelator. Thus, the training loss function of the decorrelator is given by

$$L(\theta_p, \theta_a) = L_n(\theta_p, \theta_a) + \rho_t(\max\{0, \frac{1}{n} \sum_{i=1}^{n} d(g(x_{(i)}; \theta_p), x_{(i)}) - D\})^2, \quad (23)$$

where $\rho_t$ is a penalty coefficient which increases with the number of iterations $t$. The added penalty consists of a penalty parameter $\rho$ multiplied by a measure of violation of the constraint. This measure of violation is non-zero when the constraint is violated. Otherwise, it is zero.

### D.3 EXPERIMENT SETUP

We use synthetic data generated by Gaussian mixture model as our first attempt to evaluate the performance of the learned GAPF schemes. Each dataset contains $20K$ training samples and $2K$ test samples. Each data entry is sampled from an independent multi-dimensional Gaussian mixture model. We consider two categories of synthetic datasets with $P(S = 1)$ equal to 0.75 and 0.5, respectively. Both the decorrelator and the adversary in the GAPF framework are trained on Tensor-flow (Abadi et al., 2016) using Adam optimizer with a learning rate of 0.005 and a minibatch size of 1000. The distortion constraint is enforced by the penalty method as detailed in supplement C (see equation 10).

## E   GAPF ARCHITECTURE FOR GENKI AND HAR DATASETS

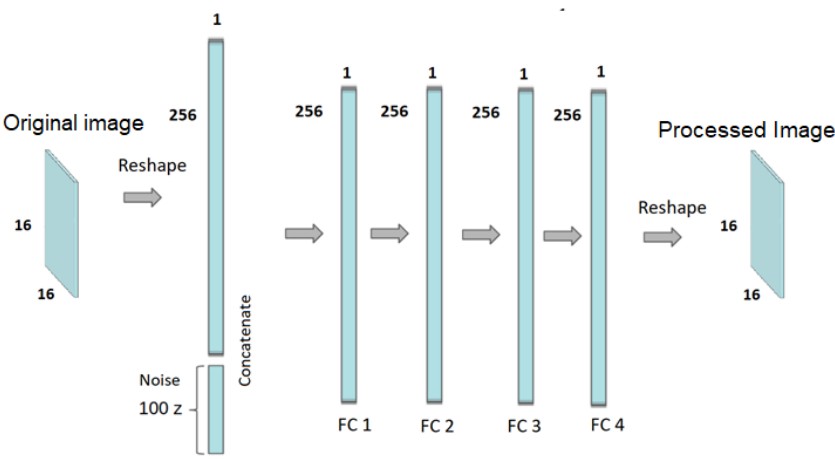

Figure 7: Feedforward neural network decorrelator

The FNND is modeled by a four-layer feedforward neural network. We first reshape each image to a vector ($256 \times 1$), and then concatenate it with a $100 \times 1$ Gaussian random noise vector. Each entry in the noise vector is sampled independently from a standard Gaussian distribution. We feed the entire vector to a four-layer fully connected (FC) neural network. Each layer has 256 neurons with a leaky ReLU activation function. Finally, we reshape the output of the last layer to a $16 \times 16$ image. To model the TCNND, we first generate a $100 \times 1$ Gaussian random vector and use a linear projection to map the noise vector to a $4 \times 4 \times 256$ feature tensor. The feature tensor is then fed to an initial transposed convolution layer (DeCONV) with 128 filters (filter size $3 \times 3$, stride 2) and a ReLU activation, followed by another DeCONV layer with 1 filter (filter size $3 \times 3$, stride 2) and a tanh activation. The output of the DeCONV layer is added to the original image to generate the processed data. For both decorrelators, we add batch normalization (Ioffe & Szegedy, 2015) on each hidden layer to prevent covariance shift and help gradients to flow. We model the adversary using convolutional neural networks (CNNs). This architecture outperforms most of other models for image classification (Krizhevsky et al., 2012; Szegedy et al., 2015).

Figure 9 illustrates the architecture of the adversary. The processed images are fed to two convolution layers (CONV) whose sizes are $3 \times 3 \times 32$ and $3 \times 3 \times 64$, respectively. Each convolution layer is followed by ReLU activation and batch normalization. The output of each convolution layer is fed to a $2 \times 2$ maxpool layer (POOL) to extract features for classification. The second maxpool layer is followed by two fully connected layers, each contains 1024 neurons with a batch normalization and a ReLU activation. Finally, the output of the last fully connected layer is mapped to the output layer, which contains two neurons capturing the belief of the subject being a male or a female.

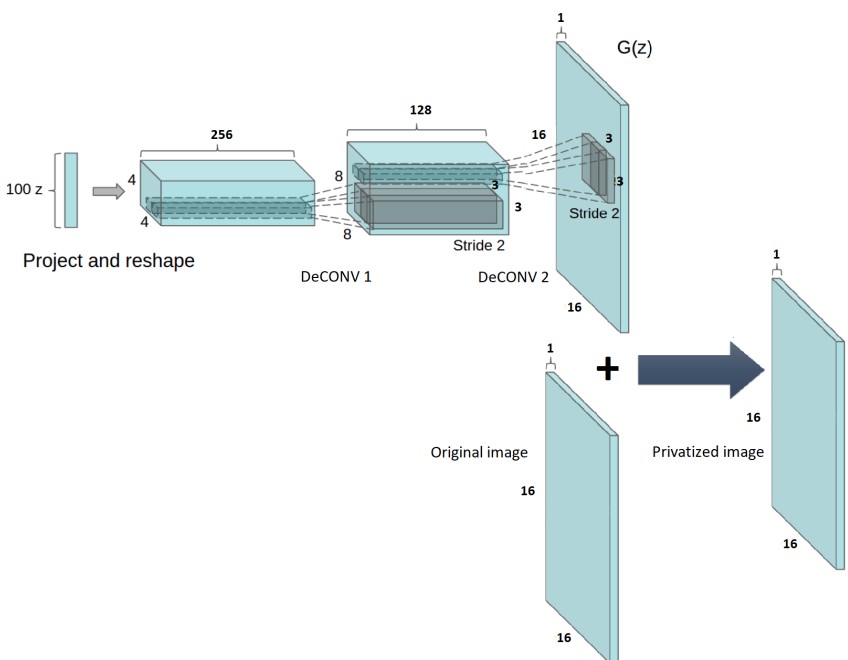

Figure 8: Transposed convolution neural network decorrelator

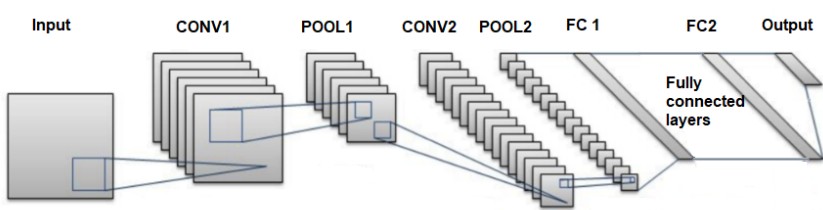

Figure 9: Convolutional neural network adversary

For the HAR dataset, We first concatenate the original data with a $100 \times 1$ Gaussian random noise vector. We then feed the entire $661 \times 1$ vector to a Feed Forward neural network with three hidden fully connected (FC) layers. Each hidden layer has $512$ neurons with a leaky ReLU activation. Finally, we use another FC layer with $561$ neurons to generate the processed data. For the adversary, we use a five-layer feedforward neural network. The hidden layers have $512, 512, 256$, and $128$ neurons with leaky ReLU activation, respectively. The output of the last hidden layer is mapped to the output layer, which contains 30 neurons capturing the belief of the subject's identity. For both decorrelator and adversary, we add a batch normalization after the output of each hidden layer.

## F  MUTUAL INFORMATION ESTIMATION

Our GAPF framework offers a scalable way to find a (local) equilibrium in the constrained min-max optimization, under certain attacks (e.g. attacks based on a neural network). Yet the privatized data, through our approach, should be immune to any general attacks and ultimately achieving the goal of decreasing the correlation between the privatized data and the sensitive labels. Therefore we use the estimated mutual information to certify that the sensitive data indeed is protected via our framework.

We use the nearest $k$-th neighbor method(Kraskov et al., 2004) to estimate the entropy $\hat{H}$ given by

$$\hat{H}(\hat{X}) = \psi(N) - \psi(k) + \log(c_d) + \frac{d}{N} \sum_{i=1}^{N} \log r_i \qquad (24)$$

where $r_i$ is the distance of the $i$-th sample $\hat{x}_i$ to its $k$-th nearest neighbor, $\psi$ is the digamma function, $c_d = \frac{\pi^{d/2}}{\Gamma(1+d/2)}$ in Euclidean norm, and $N$ is the number of samples. Notice that $\hat{X}$ is learned representation and $S$ is the sensitive variable. Then, we calculate the mutual information using $\hat{I}(\hat{X}; S) = \hat{H}(\hat{X}) - \hat{H}(\hat{X}|S)$

For a binary sensitive variable, we can simplify the empirical MI to

$$\hat{I}(\hat{X}; S) = \hat{H}(\hat{X}) - \big(P(S=1)\hat{H}(\hat{X}|S=1) + P(S=0)\hat{H}(\hat{X}|S=0)\big), \tag{25}$$

where $P(S=1)$ and $P(S=0)$ can be approximated by the empirical probability.

One noteworthy difficulty is that $\hat{X}$ usually lives in high dimensions (e.g. each image has 256 dimensions in GENKI dataset) which is almost impossible to calculate the empirical entropy based on raw data due to the sample complexity. Thus, we train a neural network that classifies the sensitive variable from the learned data representation to reduce the dimension of the data. We choose the layer before the softmax outputs (denoted by $\hat{X}_f$) to be the feature embedding that has a much lower dimension than original $\hat{X}$ and also captures the information about the sensitive variable. We use $\hat{X}_f$ as a surrogate of $\hat{X}$ for estimating the entropy. The resulting approximate MI is

$$\begin{aligned}\hat{I}(\hat{X}_f; S) &= \hat{H}(\hat{X}_f) - \hat{H}(\hat{X}_f|S) \\ &= \hat{H}(\hat{X}_f) - \big(P(S=1)\hat{H}(\hat{X}_f|S=1) + P(S=0)\hat{H}(\hat{X}_f|S=0)\big).\end{aligned}$$

Following the same manner, the MI between the learned representation $\hat{X}$ and the label $Y$ is approximated by $\hat{I}(\hat{X}_f; Y)$, where $\hat{X}_f$ is the feature embedding that represents a privatized image $\hat{X}$.

For the GENKI dataset, we construct a CNN initialized by two conv blocks, then followed by two fully connected (FC) layers, and lastly ended with two neurons having the softmax activations. In each conv block, we have a convolution layer consisting of filters with the size equals $3 \times 3$ and the stride equals 1, a $2 \times 2$ max-pooling layer with the stride equals 2, and a ReLU activation function. Those two conv blocks have 32 and 64 filters respectively. We flatten out the output of second conv block yielding a 256 dimension vector. The extracted features from the second conv layers is passed through the first FC layer with batch normalization and ReLU activation to get a 8-dimensional vector, followed with the second FC layer to output a 2 dimensional vector that applied with the softmax function. The aforementioned 8-dimensional vector is the feature embedding vector $\hat{X}_f$ in our empirical MI estimation.

Estimating mutual information for HAR dataset has a slightly different challenge, as the alphabetic size of values that the sensitive label (i.e. identity) can take is 30. Thus, it requires at least 30 neurons prior to the output layer of the corresponding classification task. In fact we pose 128 neurons before the final softmax output layer in order to get a reasonably good classification accuracy. Using the 128-dimensional vector as our feature embedding to calculate mutual information is almost impossible due to the curse of dimensionality. Therefore, we apply Principal Component Analysis (PCA), shown in Figure 10, and pick the first 12 components to circumvent this issue. The resulting 12-dimensional vector is considered to be an approximate feature embedding that encapsulates the major information of the processed data.

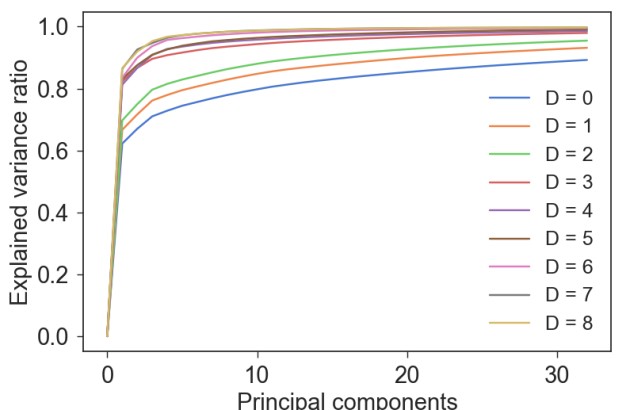

(a) Top 32 principal components out of the 561 features with different distortion $D$

Figure 10: PCA for processed data in HAR

