# OpenReview forum: "Generative Adversarial Models for Learning Private and Fair Representations"
_ICLR.cc/2019/Conference_

### Official Review · AnonReviewer1 · 2018-11-01
**Formalization of data driven GAN driven fairness methods.**

**Rating:** 7
**Confidence:** 3

**Review:**

The authors describe a framework of how to learn a "fair" (demographic parity) representation that can be used to train certain classifiers, in their case facial expression and activity recognition. The method describes an adversarial framework with a constraint that bounds the distortion of the learned representation compared to the original input.

Clarity:
The paper is well written and easy to follow. The appendix is rather extensive though and contains some important parts of the paper, though the paper can be understood w/o it.

I didn't quite follow Sec 3. It is a bit sparse on the details and the final conclusion isn't entirely clear. It also isn't clear to me how general the conclusions drawn from the Gaussian mixture model are for more complex cases.

Novelty:
Adversarial fairness methods are not new, but in my opinion the authors do a good job of summarizing the literature and formalizing the problem. I am not fully familiar with the space to judge if this is enough novelty.

Using the distortion constraint is interesting and seems to work according to the experiments. Generally though, I think that distortion can be a very restrictive constraint. One could imagine representations with a very high distortion (e.g. by completely removing the sensitive attribute) and predictive qualities equivalent to the original representation. Some further discussion of this would be good.

Experiments:
The experiments are somewhat limited, but show the expected correlations (e.g. distortion vs predictiveness).

Overall, I do believe that this work is in the right direction in this more and more popular area of great importance. I also think that contributions compared to other works could be made more clear, as well as additional experiments and discussions of the shortcomings of this approach may be added.

---

> ### Author Response · Authors · 2018-11-09
> **Authors' Response Continued**
>
>
> 4. Even though we learn our randomized decorrelation neural networks by training against a specific adversarial neural network, the learned decorrelation scheme performs well when evaluated against unseen (more complex) adversarial architectures. To prove this point, we show that the mutual information (MI) between the learned representations and the sensitive attribute is sufficiently small. A sufficiently small MI implies that no attacker (regardless of their computational power) can reliably learn the sensitive attribute from the learned representation (from Fano’s inequality [2]). This is again a novelty that didn't appear in prior works.
>
> 5. While prior works have used a classical, non-generative auto-encoder type architecture for the creation of the fair/censored representations, we harness the power of generative models which have the capability to not only compress the data in certain ways but to also cleverly inject noise where it matters (see Figure 8 and 9 in Appendix E).
>
> 6. Our set of experiments reveal that the learned representations are provably private/fair. For instance, on the GENKI dataset, we show how the gender has been stripped off by hiding mustaches, facial hair, lip color etc. (see Figure 4). At the same time, we show that the representations are still useful for other classification tasks.
>
> **Shortcomings of the distortion function**
>
> Regarding the concern about the use of distortion function, we want to point out that we are focusing on publishing datasets or meaningful representations that can be “universally” used for a variety of learning tasks which may not be known at the stage of publishing. The goal of our distortion constraint is to limit the perturbation of the data when trying to decorrelate the sensitive variable from the public variable. Thus, this distortion constraint preserves the utility of the learned representation of the data for other unknown machine learning tasks. For certain machine learning tasks, it is possible that the features related to the labels are orthogonal to the features related to the sensitive attributes. In this case, there exists a representation which completely removes the sensitive attribute with a very high distortion while the predictive qualities is still equivalent to the original representation. However, for publishing the learned representation of the data, we have to ensure this representation can also be used for a variety of learning tasks. Therefore, we impose a distortion constraint on the data to ensure that the learned representation does not deviate too much from the original data. We will fix the write up and add more discussions to this topic.
>
> **Limited experiments and some further discussions**
>
> You are correct that our simulations are limited (greyscale images and motion sensor data). We are currently working on presenting more simulation results and will post a revised version with new experimental results and detailed discussions of our contributions and shortcomings of this approach soon.
>
> References:
> [1] Kazuho Watanabe and Sumio Watanabe, Stochastic complexities of Gaussian mixtures in variational Bayesian approximation, Journal of Machine Learning Research, 7:625–644, 2006.
>
> [2] Thomas M Cover and Joy A Thomas, Elements of information theory. John Wiley & Sons, 2012.

---

> ### Author Response · Authors · 2018-11-09
> **Authors' Response**
>
> Dear AnonReviewer1,
>
> Thank you for the detailed comments and observations. We address your concerns below.
>
> **Clarity of section 3**
>
> We are currently working on rewriting section 3 to make it more accessible. This section shows that decorrelation schemes learned in a data-driven fashion against a computationally bounded adversary perform well when evaluated against a maximum a posteriori probability (MAP) adversary that has access to distributional information and knows the applied decorrelation schemes. We evaluate the learned decorrelation scheme in the following three steps:
>
> 1. We learn the decorrelation scheme in a data driven fashion using synthetic Gaussian mixture data.
> 2. We evaluate the performance of the learned decorrelation scheme under a strong adversary who has access to dataset statistics, knows the learned decorrelation scheme, and can compute the MAP decision rule.
> 3. We compare the performance of the learned scheme with the game-theoretic optimal one.
>
> The first step can be done for any dataset but the last two steps can only be done for data that we have access to its distribution. Since the distribution of a real dataset is very difficult to obtain. We assume that the public variable follows a Gaussian mixture model conditioned on the value of the sensitive variable. In this case, we can compute the game-theoretic optimal decorrelation scheme and the optimal decision rule of the strong adversary. We agree that the conclusion drawn from the Gaussian mixture model is limited and may not generalize to more complex model. But this serves as a good sanity check, especially given that Gaussian mixture models have been used in many areas [1].
>
> **Novelty of this paper**
>
> We would like to list the novelty of this paper and highlight the important contributions as follows.
>
> 1. All the relevant papers have exclusively focused on showing (via experiments) that this approach works well in practice when you design things with a particular classification task in mind (i.e., in a supervised fashion). This requires having access to additional training labels which may be unavailable during the training phase. Our paper shows (via experiments on medium-sized datasets) that this approach works even when the designer does not want to restrict their attention to one classification task (i.e., in an unsupervised fashion). Indeed, our experiments and simulations show that the learned representations work well on classification tasks that haven't been accounted for.
>
> 2. We make very precise connections between the data-driven adversarial learning framework and the game- and information-theoretic setting (which assumes that the designer has access to the joint distributions between data and sensitive attributes, and the minimax optimization is performed over all theoretically possible randomized decorrelation and adversarial learning rules). We also show how the change of the loss function in our framework leads to a variety of information theoretic adversaries with different powers. This is an important novelty because it allows us to generalize conclusions that can be made upon learning representations from a finite number of samples against a computationally bounded adversary to the more important setting of infinite samples (i.e., access to distributional information) and infinite adversarial computational power. Indeed, this is explicitly shown in Section 3 for Gaussian mixture models. Notice that this section shows that decorrelation schemes that are learned in a data-driven fashion against a computationally bounded adversary perform well when evaluated against a maximum a posteriori probability (MAP) adversary that has access to distributional information and knows the applied decorrelation schemes. Further, this shows that there is no gap between the game-theoretically optimal decorrelation schemes and the ones that are learned via a generative neural network for binary variable S.
>
> 3. Different from previous works where the objective is modeled as a weighted combination of loss functions and distortion penalty, our formulation is a minimax game subject to a "hard distortion constraint". This allows us to directly limit the amount of distortion added to learn the representation, which is crucial for preserving the utility of the learned representation. Moreover, notice that enforcing the hard distortion constraint calls for a new training process that relies on the Penalty method or Augmented Lagrangian method presented in Appendix C.

---

### Official Review · AnonReviewer3 · 2018-11-07
**Interesting direction and formulation but no enough novelty**

**Rating:** 4
**Confidence:** 3

**Review:**

This paper present an adversarial-based approach for private and fair representations. This is done by learned distortion of data that minimises the dependency on sensitive variable while the degree of distortion is constrained. This problem is important, and the analysis from game-theory and information theory perspectives is interesting. However, the approach itself is similar to Edwards & Storkey 2015, and I find the presentation of this paper confusing at a few points.

First, while both the title and abstract suggest it is about learning representation, the approach might be better considered as data-augmentation. As described a bit later: "...modifying the training data is the most appropriate and the focus of this work". This contradiction with more commonly accepted meaning of representation learning (learning abstract/high level representation of data) is confusing.

Although the authours argued this work is different from Edwards & Storkey 2015, I think they are quite similar. The presented method is almost a special case of this previous work: it seems that one can obtain this model by modifying Edwards & Storkey's model as follows (referring to the equations in Edwards & Storkey's paper): (1) removing the task (Y) dependent loss in eq. 9. (2) assume the encoder transforms X to the same data space so the decoder can be removed, so eq. 7 become equivalent to the distortion measure in this paper. There are other small differences, such as adding noise and the exact way to impose constraint, but I doubt whether the novelty is significant in this case.

Other places that are unclear include: proposition 1 -- what does "demographic parity subject to the distortion constraint" mean? demographic parity was defined earlier as complete independence on sensitive variable, so how can "complete independence" subject to a constraint? In addition, it would be helpful introduce S is binary. This information was delayed to section 3 after the cross-entropy loss that assumes binary S was presented.

Overall, I think this paper is interesting, and the analysis offers insights into related areas. However, the novelty is not enough for acceptance at ICLR, and the presentation can be improved.

---

> ### Author Response · Authors · 2018-11-09
> **Authors' Response Continued**
>
> 5. While prior works have used a classical, non-generative auto-encoder type architecture for the creation of the fair/censored representations, we harness the power of generative models which have the capability to not only compress the data in certain ways but to also cleverly inject noise where it matters. (see Figure 8 and 9 in Appendix E)
>
> 6. Our experiments reveal that the learned representations are private/fair even to humans. For instance, on the GENKI dataset, we show how the gender has been stripped off by hiding mustaches, facial hair, lip color etc. (see Figure 4). At the same time, we show that the representations are still useful for other classification tasks.
>
> **Confusion in demographic parity subject to the distortion constraint**
>
> We would like to clarify what we meant by "demographic parity subject to a distortion constraint." It is well known in the fairness community that enforcing demographic parity (or other notions of fairness) conflicts with the learning of well-calibrated classifiers ([3,4]). To circumvent this issue, we chose to "partially" decorrelate the data up to an allowed distortion. This helps in ensuring that the learned representations are useful in practice for learning good classifiers, while limiting the underlying correlations with the sensitive attributes. Our formulation implies demographic parity if the distortion budget is set to infinity (see the analysis in Appendix B, proof of proposition 1).
>
>  **Introduce S is binary**
>
> We would like to emphasize that the proposed framework is general and can be used for non-binary sensitive variable. However, in the theoretical analysis, we only consider binary sensitive variable. The analysis can be generalized to the non-binary case. Furthermore, we also consider non-binary sensitive variable in our simulation (the HAR dataset). We will fix our writeup to clarify this.
>
> With all of the above in mind, we hope we have made a case for the innovation in our work and convinced you to reevaluate your assessment of our work. We are happy to further discuss and clarify any concerns you may still have.
>
> References:
> [1] Harrison Edwards and Amos Storkey, Censoring representations with an adversary, In Proceedings of the International Conference on Learning Representations, San Juan, Puerto Rico, May 2016.
>
> [2] Thomas M Cover and Joy A Thomas, Elements of information theory, John Wiley & Sons, 2012.
>
> [3] Cynthia Dwork, Moritz Hardt, Toniann Pitassi, Omer Reingold, and Richard Zemel, Fairness through awareness. In Proceedings of the 3rd innovations in theoretical computer science conference, pp. 214–226. ACM, 2012.
>
> [4] Moritz Hardt, Eric Price, Nathan Srebro, Equality of opportunity in supervised learning. In Advances in neural information processing systems, pp. 3315–3323, 2016.

---

> ### Author Response · Authors · 2018-11-09
> **Authors' Response**
>
> Dear AnonReviewer3,
>
> Thank you for the detailed comments and observations. We are happy you found our paper and analysis interesting. We understand there is room for improvement in the write up. We are currently working on refining it; even as we do so, we respond here to your comments to address as precisely as we can.
>
> **Confusion about the term “representation learning”**
>
> We agree that the term “learning private and fair representations” might be confused with the widely studied “representation learning” problem -- which we are not tackling in this work. While our framework can be generalized to a setting in which we can learn an arbitrary representation using an encode-decode structure, we are primarily interested in learning representations of the data (of the same dimension/shape/structure) that are fair and private. Thank you for pointing this out. We will fix our writeup to clarify things.
>
> **Difference between our work and Edwards & Storkey 2015 [1]**
>
> Our work departs (quite significantly) from other related works. Here is a list of the important differences.
>
> 1. Our framework is not a special case of Edwards & Storkey 2015. For starters, our formulation is a minimax one subject to a "hard distortion constraint". Their formulation is a weighted combination of three loss functions, and zeroing out one of them (the one that measures how well you do in a given classification task of interest) does not recover our formulation because of the non-convexity/concavity of the minimax problem with respect to the decorrelator/adversary neural network parameters. The hard distortion constraint allows us to directly limit the amount of distortion added to learn the private/fair representation, which is crucial for preserving the utility of the learned representation. Moreover, notice that enforcing the hard distortion constraint calls for a new training process that relies on the Penalty method or Augmented Lagrangian method presented in Appendix C.
>
> 2. All the relevant papers have exclusively focused on showing (via experiments) that this approach works well in practice when you design things with a particular classification task in mind (i.e., in a supervised fashion). This requires having access to additional training labels which may be unavailable during the training phase. Our paper shows (via experiments on medium-sized datasets) that this approach works even when the designer does not want to restrict their attention to one classification task (i.e., in an unsupervised fashion). Indeed, our experiments and simulations show that the learned representations work well on classification tasks that haven't (at all) been accounted for in the training process.
>
> 3. We make very precise connections between the data-driven adversarial learning framework and the game- and information-theoretic setting (which assumes that the designer has access to the joint distribution between data and sensitive attributes, and the minimax optimization is performed over all theoretically possible randomized decorrelation and adversarial learning strategies). We also show how the change of the loss function in our framework leads to a variety of information-theoretic adversaries with different powers. This is an important novelty because it allows us to generalize conclusions that can be made upon learning representations from a finite number of samples and a computationally bounded adversary to the more important setting of infinite samples (i.e., access to distributional information) and infinite adversarial computational power. Indeed, this is explicitly shown in Section 3 for Gaussian mixture models. Notice that this section shows that decorrelation schemes that are learned in a data-driven fashion against a computationally bounded adversary perform well when evaluated against a maximum a posteriori probability (MAP) adversary that has access to distributional information and knows the applied decorrelation schemes. Further, this shows that there is no gap between the game-theoretically optimal decorrelation schemes and the ones that are learned via a generative neural network for binary variable S. This critical piece where one investigates what guarantees we can get against more potent adversaries is missing from prior works.
>
> 4. Even though we learn our randomized decorrelation neural networks by training against a specific adversarial neural network, the learned decorrelation scheme performs well when evaluated against unseen (more complex) adversarial architectures. To prove this point, we show that the mutual information (MI) between the learned representations and the sensitive attribute is sufficiently small. A sufficiently small MI implies that no attacker (regardless of their computational power) can reliably learn the sensitive attribute from the learned representation (from Fano’s inequality [2]). This is again a novelty that didn't appear in prior works.

---

### Official Review · AnonReviewer4 · 2018-11-17
**Unconvincing newness,  but a good GAN model to understand Private Presententation Learning,**

**Rating:** 4
**Confidence:** 3

**Review:**

    The paper authors provide a good overview of the related work to Private/Fair Representation Learning (PRL). Well written, The theoretical approach is extensively explained and the first sections of the paper are easy to follow. The authors demonstrate the model performance on or the GMM, the comparison between theoretical and data driven performance is a good case study to understand the PRL.

We usually expect to see related work in the first sections, in this case it's has been put just before the conclusion. It can be still justified by the need o introduce the  PRL concepts before comparing with other works.
The GMM study case is interesting, but incorporates strong assumptions. Moreover, for a 4 or 8 dimensional GM, 20K data points are more than enough to infer the correct parameter. It would have been more useful if it was used to comapre between the mentioned methods in "Related Work".

There seems to be important parts of the paper that has been put in the appendices: how to solve the constrained problem, Algorithm.... Similarly, some technical details were expanded in the paper body (Network structure).

The authors mentioned the similarities with other works and their model choices that set theirs apart from other. Yet, the paper doesn't provide performance ( accuracy, MI) comparison to other works. There seems to be a strong similarity with Censoring representations with an adversary, Harrison Edwards and Amos Storke (link: https://arxiv.org/abs/1511.05897). Difference : distortion instead of H divergence, non-generative autoencoders.

Consequently, I question the novelty of the paper's contribution. Without extensive comparison with other methods and especially to similar ones mentioned in the related work, there is little to say about the "state-of-the-artness". Yet, it is important to acknowledge the visible effort behind the paper and how the author(s) managed to leverage the simplicity and power of GANs.

On a lighter note:
A)- the paper mention "state-of-the-art CNNs, state-of-the-art entropy estimators, MI, generative models", for the Machine Learning community, many of these elements have been around for a while now.
B)- "Observe that the hard constraint in equation 2 makes our minimax problem different from what is extensively studied in the machine learning community": I would argue it's not an objective statement.

---

> ### Author Response · Authors · 2018-11-27
> **Authors' Response**
>
> Dear AnonReviewer4,
>
> Thank you for the detailed comments and feedback. We would like to address your concerns below.
>
> **Related work in the first section**
>
> Thank you for the comment. We have moved the related work section to the introduction to make it more accessible. We have also provided a more detailed literature review section in Appendix A.
>
> **GMM study case**
>
> For the multi-dimensional Gaussian mixture data model, we derive game-theoretically optimal decorrelation schemes and compare them with those that are directly learned in a data-driven fashion to show that the gap between theory and practice is negligible. The goal here is not to show how well we can learn a machine learning model from Gaussian mixture data. Our goal is to provide formal verification of how fair/private schemes that are learned by competing against computational adversaries with a fixed architecture generalize against adversaries with more complex architecture. If we have a smaller number of samples, we expect to see the performance of the learned decorrelation scheme degrades. We have included the numerical results of the 32-dimensional Gaussian mixture data in Section 3 of the revision. We observe that the learned decorrelation scheme performs as well as the game-theoretically optimal one for the 32-dimensional Gaussian mixture data.
>
> **Important parts in the appendix**
>
> We have included more descriptions of the constrained minimax game and how to enforce the distortion constraint in the alternate minimax algorithm in Section 2. We have also included the theoretical results and a detailed description of the GMM model in Section 3.
>
> **Comparison with other papers**
>
> Existing relevant works (such as Edwards & Storkey 2016) have two limitations: (1) they require the designer to have a specific classification task in mind at training time, and (2) they only give empirical guarantees against computationally finite adversaries. Indeed, it is natural and important to ask: (1) how can we obtain representations that work well without having a specific learning task in mind at training time, and (2) how well do the learned representations perform against more powerful adversaries? Our work addresses both limitations. To address the first, we introduce a constrained minimax formulation that ensures that the utility is preserved by enforcing a distortion constraint. To address the second, we show (in Theorem 1, Corollary 1, and Proposition 1) how our framework recovers an array of (operationally motivated) information-theoretic notions of information leakage. We critically leverage this connection to show that representations learned in a data-driven fashion against a finite adversary are as good as representations that are game-theoretically optimal. We further introduce a systematic approach using mutual-information estimation to certify that the learned representations will perform well against unseen, more complex adversaries. We encourage the reviewer to take a look at our response to AnonReviewer3, where we listed our contributions in great detail.
>
> **On a lighter note**
>
> Thank you for the feedback. We have included these comments in the revision.
> A: We have removed “state-of-the-art” in our paper.
> B: To the best of our knowledge, our approach is the first to apply “distortion constraints” to learn a fair/private representation of datasets that can be used for a variety of learning tasks. We have changed “in the machine learning community” to “in previous works” to avoid the confusion.

---

### Public Comment · (anonymous) · 2018-11-11
**Relevant References**

Hi, just pointing out some related papers.

1. Xu, Depeng, Shuhan Yuan, Lu Zhang, and Xintao Wu. "FairGAN: Fairness-aware Generative Adversarial Networks." arXiv preprint arXiv:1805.11202 (2018).
2. Sattigeri, Prasanna, Samuel C. Hoffman, Vijil Chenthamarakshan, and Kush R. Varshney. "Fairness GAN." arXiv preprint arXiv:1805.09910 (2018).

---

> ### Author Response · Authors · 2018-11-13
> **Response to Relevant References**
>
> Dear Anonymous,
>
> Thank you very much for these references. We will add them to our related work in the revised version.
>
> The two papers you mentioned focus on generating synthetic non-sensitive attributes and labels which ensure fairness while preserving the utility of the data (predicting the label). The synthetic data is generated by a conditional generative adversarial network (GAN) which generates the non-sensitive attributes-label pair given the noise variable and the sensitive attribute. The utility is preserved by generating data that is very similar to the original data. To ensure fairness, the generator generates data samples such that an auxiliary classifier (discriminator) trained to predict the sensitive attribute from the synthetic data performs as poorly as possible.
>
> FairGAN uses two discriminators: one discriminates fake / real non-sensitive attributes-label pair and another discriminates generated data from different sensitive groups. This model ensures demographic parity while preserving data utility (predicting the label). The problem is formulated as an unconstrained minimax game in which the empirical loss function is formulated by a weighted sum of the loss functions of the two discriminators. Fairness GAN is similar to FairGAN. The goal here is to develop a conditional GAN-based model to ensure demographic parity or equality of opportunity in the system by learning to generate a fairer dataset. The authors consider both demographic parity and equality of opportunity as fairness metric. They also formulate the problem as an unconstrained minimax game between the discriminator and the generator. To ensure utility, the Fairness GAN uses three pairs of losses which make sure that the generated non-sensitive attributes-label pair, the non-sensitive attributes alone as well as the non-sensitive attributes conditional on the sensitive attributes to be very similar to the original data. To enforce fairness, they include a pair of losses to encourage either demographic parity or equality of opportunity.
>
> The methods presented in these papers are very different from our method. First, we are focusing on creating representations of the data for a variety of learning tasks. Second, we use a generative model to cleverly injecting noise where it matters to ensure privacy/fairness rather than generate a fairer synthetic dataset. Third, we consider a constrained minimax game in which we use a distortion constraint to preserve the utility of the learned representation for a variety of learning tasks rather than focusing on a particular label. Fourth, we make precise connections between the data-driven adversarial learning framework and the game- and information-theoretic setting (with knowledge of dataset statistics) and show how the change of the loss function in our framework leads to a variety of information-theoretic adversaries with different powers. Furthermore, we use simulations on Gaussian mixture models to show that the learned representations from a finite number of samples and a computationally bounded adversary (neural networks) performs as good as a representation created by the game-theoretic optimal mechanism which assumes knowledge of dataset statistics and infinite adversarial computational power. Finally, we propose using mutual information estimators to verify that no adversary (regardless of their computational power) can reliably learn the sensitive attribute from the learned representation. We encourage the reader to read the detailed list of contributions below where we attempt to make this clear to ICLR reviewers. There are different ways for enforcing fairness, and our work presents a framework that aids in achieving this goal. More work is needed to be done in this area.
>
> We thank you again for your comment and bringing these references to our attention.

---

### Author Response · Authors · 2018-11-27
**Summary of changes in the revised paper**

We thank all the reviewers and readers for their feedback. We have updated our paper on OpenReview. We list below the major changes we have made to the paper.

1. We have rewritten “our contributions” subsection to highlight our main contributions.

2. We have moved the “related work” section to the introduction and highlighted the major differences between our work and other related work. We have also included the references on using GAN to generate synthetic fair dataset. A detailed literature review is provided in Appendix A.

3. We have rewritten Theorem 1 and added Corollary 1 to incorporate a general alpha-loss function, which provides a continuous interpolation between a hard decision adversary (using 0-1 loss) and a soft decision adversary (using log-loss). We have also rewritten proposition 1 and its proof to provide a clearer explanation of how GAPF enforces demographic parity.

4. We have included more detailed explanation of the constrained minimax game and how to enforce the distortion constraint in section 2.

5. We have rewritten section 3 to make our results of the Gaussian mixture model more accessible. We have also added numerical results of the 32-dimensional Gaussian mixture model.

6. We have moved the technical details of the decorrelator and adversary network architecture in section 4 to Appendix E. We have also included more simulation results of the GENKI dataset.
    6.1 We have added the false error rate of the facial expression classifier learned from the private/fair representation to show that GAPF enforces fairness (Table 1 and 2).
    6.2 We have included the mutual information estimation of the Transposed Convolution Neural Network Decorrelator.
    6.3 We have plotted the gender classification accuracy as a function of the expression classification accuracy to illustrate the tradeoff between enforcing privacy/fairness and preserving the utility of the representation.

7. We have fixed some grammatical errors and typos. We have also corrected some technical terms to make the paper more accessible.

---

### Meta-Review · Area_Chair1 · 2018-12-17
**Not ready for publication at ICLR**

**Confidence:** 5
**Recommendation:** Reject

**Metareview:**

While there was some support for the ideas presented, the majority of the reviewers did not think the submission was ready for presentation at ICLR. Concerns raised included that the experiments needed more work, and the paper needs to do a better job of distinguishing the contributions beyond those of past work.